# Comparing artificial intelligence and human coaching goal attainment efficacy

**Nicky Terblanche**[1]*, **Joanna Molyn**[2], **Erik de Haan**[3,4], **Viktor O. Nilsson**[3]

**1** University of Stellenbosch Business School, Cape Town, South Africa, **2** University of Oxford Brookes, Oxford, United Kingdom, **3** Ashridge Centre for Coaching, Hult International Business School, Berkhamsted (Herts.), United Kingdom, **4** VU University Amsterdam, Amsterdam, The Netherlands

* nickyt@usb.ac.za,

## Abstract

The history of artificial intelligence (AI) is filled with hype and inflated expectations. Notwithstanding, AI is finding its way into numerous aspects of humanity including the fast-growing helping profession of coaching. Coaching has been shown to be efficacious in a variety of human development facets. The application of AI in a narrow, specific area of coaching has also been shown to work. What remains uncertain, is how the two compare. In this paper we compare two equivalent longitudinal randomised control trial studies that measured the increase in clients' goal attainment as a result of having received coaching over a 10-month period. The first study involved human coaches and the replication study used an AI chatbot coach. In both studies, human coaches and the AI coach were significantly more effective in helping clients reach their goals compared to the two control groups. Surprisingly however, the AI coach was as effective as human coaches at the end of the trials. We interpret this result using AI and goal theory and present three significant implications: AI coaching could be scaled to democratize coaching; AI coaching could grow the demand for human coaching; and AI could replace human coaches who use simplistic, model-based coaching approaches. At present, AI's lack of empathy and emotional intelligence make human coaches irreplicable. However, understanding the efficacy of AI coaching relative to human coaching may promote the focused use of AI, to the significant benefit of society.

## Introduction

Since its inception in the 1950s, artificial intelligence (AI) has seen several periods of growth and decline, casting doubt on its actual versus claimed efficacy [1]. Lately, renewed interest in AI has led to numerous novel applications of this technology, including in healthcare and helping professions such as psychology and coaching [2–4]. In this paper, coaching is defined as a one-on-one structured conversation between a coach and client with the aim of facilitating sustainable change for the individual and potentially other stakeholders [5]. Coaching is a late entrant to the application of AI, and AI's role and efficacy in coaching remain largely under-researched.

Coaching is an important helping profession. It is a fast-growing multi-billion dollar per year industry [6] and has grown substantially both in practice and research in the last decade

**Data Availability Statement:** Two data files are included as Supplementary files as part of this submission.

**Funding:** The author(s) received no specific funding for this work.

**Competing interests:** The authors have declared that no competing interests exist.

[7]. Numerous coaching meta-studies have made a clear case for its efficacy [8–13]. There is a strong link between successful coaching outcomes and the relationship and bond between the coach and client with convincing evidence that the coach-client relationship is the most significant factor in coaching success [14–16]. Nevertheless, the current limitations of AI, especially relating to true human intelligence and emotions [17], cast doubt over the ability of an AI coach to currently compete with a human coach.

Recent studies on the application of AI in psychology, however, have suggested that AI could be effective in certain domains of promoting human wellbeing. Fulmer et al. [18], for example, used an AI agent based on cognitive behavioral therapy (CBT) to reduce self-identified symptoms of depression and anxiety in college students. They concluded that AI could serve as a cost-effective and accessible therapeutic agent. Greer et al. [19] found that young adult cancer patients had reduced anxiety compared to a control group after using a positive psychology-based AI coach for four weeks. These findings suggest that, while AI lacks true human intelligence and emotions, positive outcomes are possible even in practices that have traditionally relied on a strong human connection. This might potentially also be the case for coaching.

One of the primary focal areas of coaching and what sets it apart from other helping professions, is assisting clients with goal attainment [20, 21]. Understanding the efficacy of AI coaching compared to human coaching in the domain of goal attainment therefore seems like a reasonable starting point for AI coaching research. This leads us to ask the following research question: *In a similar setting, how does AI coaching compare to human coaching efficacy in relation to client goal attainment*?

In this paper, we investigate this question by presenting a comparison of the two studies on goal attainment coaching: the first involving human coaches and the second an AI coach. We interpret the results in terms of the current state of AI and goal theory. We also discuss the way these results may pave the way for aspects of coaching to be made more widely available and the implication for coaches and the coaching industry. Given the continued growth of coaching as a helping profession and its proven efficacy, understanding how AI could play a role in scaling and democratizing this service is an important research area.

## Current capabilities of artificial intelligence

AI has seen several false starts mostly because of exaggerated claims of progress and ability that inevitably led to disappointment. An example is Marvin Minsky, the father of AI who back in 1967 stated that "Within a generation . . . the problem of creating artificial intelligence will substantially be solved" [22]. AI has experienced a few "winters" where these types of exaggerations led to withdrawal of funding and the collapse of interest in AI research and development [1]. However, the recent resurgence in AI interest appears to be more sustainable as AI is focused on specific specialist areas in line with current AI capabilities, and shows promise in areas such as decision-making processes [23].

AI is defined as "the broad collection of technologies, such as computer vision, language processing, robotics, robotic process automation and virtual agents that are able to mimic cognitive human functions" [24 p4]. However, in order to understand AI's realistic capabilities, it is important to distinguish between three types of AI: (i) Artificial narrow intelligence (ANI) refers to systems that can perform a specific task in a narrow context, such as a self-driving car; (ii) Artificial general intelligence (AGI) refers to systems that have intelligence similar to human intelligence; and (iii) Artificial super intelligence (ASI) refers to systems that can outperform human intelligence [25–27]. AGI and ASI do not currently exist and by acknowledging this fact, it creates a more realistic picture of what AI and specifically ANI can accomplish [28].

For the foreseeable future AI entities will remain unconscious machines that can at best support humans in complex, specific tasks [17]. This implies that ANI systems will be highly specialised and skilled in specific tasks and may even outperform humans in these narrow focus areas [29]. Perhaps instead of waiting for true AGI, multiple narrow AI applications could be interconnected to collaboratively perform tasks in a synergistic manner, possibly with utility beyond what a singly ANI application could do [17]. AI is not yet poised to completely replace humans; however, the improved ability of AI and its increased use in the helping professions suggest we need to investigate more closely the relationship between AI and human interaction. In the highly human-centric context of coaching, the human-AI relationship becomes critical.

## Human-AI interactions and relationships in the context of coaching

An indication of the growing importance of human-AI interaction is the emergence of studies augmenting current human-computer interaction (HCI) theory dedicated to human-centered AI (HCAI) [30] and human-AI interaction (HAII) [31]. The focus of AI development seems to be shifting away from pure scientific and academic exploration to useful applications that also consider human factors [30]. Human factors include the creation of AI systems that have social benefits and consider the ethical implications of AI. It also includes the consideration of the role of humans in the AI ecosystem and awareness of the need for a more human-centred approach [32, 33]. The advancement of AI, combined with the focus on placing humans at the centre, have led to new development of AI roles, ranging from being purely assistive to helping with team collaboration [34, 35]. The fast-growing area of AI-assisted decision-making, for example, requires clear boundaries on human versus AI authority and accountability. This is observed in the healthcare industry context, where decisions on patient care and diagnosis can have life or death consequences. As healthcare professionals team up with AI, there is a real danger that the "third wheel" effect (additional, potentially redundant or confusing opinions) may decrease combined human and AI effectiveness [23].

The present study is not focused on augmented human plus AI interaction since the AI coach used operates autonomously from a human coach. However, the AI coach's sole task is to interact with (coach) a human client. Therefore, the interaction and especially the relationship between the AI and human remain important. There are several suggested ways to create AI coaches that focus on strong human-AI relationships.

Of primary concern is the need for the AI to have social ability, demonstrate credibility and context awareness and be proactive in assisting clients [36]. It is also important that the AI coach strives to embody the aspects that make human coaching effective, including demonstrating trust, empathy, transparency, predictability, reliability, ability, benevolence, and integrity. To create a strong AI-human relationship these aspects can be operationalized as suggested in Table 1 (see Terblanche [4] for a detailed discussion).

## Potential benefits and ethical challenges in AI coaching

The application of AI in the helping professions and in coaching specifically holds numerous potential benefits. In the related field of psychology AI offers new modes of treatment, the ability to reach currently excluded populations, improve patient response and free up limited resources such as highly trained psychologists [51]. These same advantages apply to coaching.

The benefits of coaching are well researched and several meta-studies have shown that coaching can help people with various aspects including: performance and skills; wellbeing; coping; work attitudes; goal-directed self-regulation; improved work/life balance; psychological and social competencies; self-awareness and assertiveness, increased confidence;

**Table 1. AI design practices to support strong coach-coachee relationships.**

| Coach attribute | AI design consideration |
|---|---|
| **Trust** | • Avoid the 'uncanny valley' effect [37]<br>• Communicate data privacy agreement [38]<br>• Create consistent AI personality [39]<br>• Reduce security and privacy concerns [40] |
| **Empathy** | • Use a human name and human-like conversational cues [41]<br>• Remember user's likes, dislikes and preferences across sessions [40] |
| **Transparency** | • Reveal non-humanness [42]<br>• Practice self-disclosure [43]<br>• Showcase purpose and ethical standards [44] |
| **Predictability** | • State possible behaviour change due to continuous learning [42]<br>• Find a balance between a predictable personality and sufficient human-like variation [45]<br>• Use conversational context in interactions [46] |
| **Reliability** | • Fail gracefully [42]<br>• Monitor chatbot performance and reliability [42]<br>• Provide confirmation messages [40] |
| **Ability** | • Use established theoretical models (e.g. goal attainment) [47, 48]<br>• Use personalisation and avoid generic responses [49] |
| **Benevolence** | • Communicate positive intent [42]<br>• Demonstrate a positive attitude and mood [40] |
| **Integrity** | • Clearly communicate limitations [42]<br>• Clarify purpose in the introductory message [50] |

developing relationships, networks and interpersonal skills; adapting to change more effectively; helping to set and achieve goals; role clarity; and changing behaviors [9, 10, 13]. However, not everyone has access to a coach, especially in less affluent societies. In Africa, for example, the average cost of an organisational coach is approximately 100 USD per session, which puts it out of reach of many [52]. The problem is not only cost. There is a dire shortage of skilled coaches in many parts of the world. Of the more than 40,000 coaches registered with the International Coaching Federation, fewer than 2,000 are in Africa [53]. It seems that currently most of humanity is excluded from the benefits of professional coaching, even though there are calls for coaching to be viewed as a social process that could benefit currently marginalised groups [54]. AI potentially holds the key to expanding the reach of coaching. The ability of AI to scale and provide basic coaching services at a vastly reduced cost could overcome these current limitations, possibly democratizing coaching to the significant benefit of society.

The use of AI in coaching raises ethical concerns. These include prevention of harm, lack of guidance on developing ethical AI, respect and protection of client autonomy, transparency in the use of algorithms, bias, and data ownership [4, 51]. For AI coaching to be widely accepted and trusted, these ethical challenges must be addressed by stakeholders [36].

## Goal theory

An important theoretical foundation of this paper is goal theory as applied in coaching. Goal theory is well established and widely used due to a history of empirical research and application. It is in essence an approach explaining the need to establish goals as an intrinsic motivation where a relationship exists between goal difficulty, level of performance, and effort involved [55]. Goal theory is supported by five principles regarding goal setting: clarity (specific and clear); challenge (sufficiently difficult); commitment (buy-in from onset); feedback (regular stock-taking on progress); and complexity (not too complex) [56]. Goals are "internal representations of desired states or outcomes" [57 p388]. Goal setting and attainment have been shown to have a positive effect on workplace performance [56]. Goal attainment has also been linked to positive emotions and increased wellbeing [58–60].

Various factors influence peoples' goal attainment success. A study by Klein and Fishbach [61] showed that disrupting the expectations of goal attainment may lead to reduced satisfaction and lower goal evaluation, even though the goal is eventually achieved. Other factors that could influence goal attainment include the experience of power whereby people who feel less powerful are less motivated to reach their goals when the goal seems far away [62]. People from cultures where personal honor is important may also delay their goal pursuit if they receive a threat to their moral reputation, such as being called a liar [63].

Certain actions can enhance goal attainment, including the writing down of goals, measuring goals and having specific time frames attached to them, and making a public commitment to someone regarding the goal [55]. There are various types of goals, for example, proximal and distal goals [64].

Goal theory is used extensively in coaching as an underlying mechanism to facilitate self-regulation [64]. During coaching, individuals with the help of their coach set goals, develop and execute action plans, monitor progress and change either goals or action plans based on feedback and progress [65]. Coaching in particular provides the monitoring function that helps to translate goals into actions, which in turn leads to progress [66].

Several empirical studies have shown that coaching is effective in improving goal attainment. In a randomized control trial (RCT) study, Grant et al. [67] found that four coaching sessions over a 10-week period led the intervention group to a significantly higher level of goal attainment compared to the control group. Zimmerman and Antoni [68] analyzed 33 coaching dyads using longitudinal multilevel analyses and found that clients experienced increased goal attainment. Losch et al. [69] compared individual coaching, self-coaching and group training and found that individual coaching was effective and superior in helping leaders achieve their goals.

As goal theory is intrinsic to coaching and since coaching has been shown to improve goal attainment, it emerged as an appropriate theory to investigate an impact of AI coaching on goal attainment in the present study.

## Coach maturity

With the coaching industry growing rapidly, a diverse range of people are attracted to become coaches for reasons ranging from the promise of increased freedom, balanced lifestyle, self-control and a reprieve from corporate politics, bureaucracy and pressure [70]. Coaching is an unregulated industry, which implies that coaches enter the profession with various levels of training and experience [71], ranging from no training at all to post-graduate degrees [72]. The result is that coaches practice at different levels of coach maturity [73]. The notion of coach maturity is an important consideration given that AI has taken over the jobs of some people [74], suggesting that humans coaches who operate at a low level of complexity may be rivalled by AI coaching.

Megginson and Clutterbuck [73] distinguished among four levels of coach maturity. At the lowest level coaches follow a "models-based" approach where they are typically more interested in following a set, mechanistic process rather than exploring the complexities of the client's world. They are "doing coaching to the client". This type of coaching is typical of novice coaches who rely on the coaching skills and techniques they had been trained in initially [72]. On the second level, "process-based" coaches follow a slightly more flexible approach using an expanded but still limited set of tools and techniques. They are "doing coaching with the client". On the third level, "philosophy-based" coaches apply a broader mind-set to the client's situation and practice reflection before and after coaching sessions. The top level, "systemic eclectic" is acquired through much experience and allows a coach to exhibit a sensitive, intelligent approach to the client situation and utilize the most appropriate approach give the context. They are part of the system in which the coaching occurs. This coach maturity model was

summarized by Drake [75 p143]: ". . .as novices they learn the rules, as intermediates they break the rules, as masters they change the rules and as artisans they transcend the rules".

This coach maturity model is testimony to how humans can integrate knowledge and apply learning across domains, allowing navigation of complex situations. While AI is currently incapable of this, the fact that ANI can perform specific tasks on a level of human competency and beyond [29] suggests that the lowest level of coach maturity (models-based) is potentially within the ability of a well-designed narrow AI system.

## Methodology

The two studies we compare were both longitudinal RCT designs. The studies were designed with the CONSORT guidelines for RCT research in mind and these guidelines were adhered to as far as possible [76]. Both studies were conducted over ten months with different groups of participants. Study 1 ran from October 2017 to July 2018 and consisted of a control group and a human coach group where participants received coaching from a human coach. Study 2 ran from November 2019 to August 2020 and consisted of a control group and an intervention group where participants received coaching from an AI chatbot coach. The second study was conducted after the first one because the AI coach was only created in 2019 after the completion of the first study. The same data collection instruments were used in both studies.

The research was approved by the ethics committee of a London-based University, project reference UREC/19.1.5.6. Written informed consent was obtained from all participants in both studies as per the requirements of the ethics approval.

## Participants

Participants in both studies were recruited via email from a business school in the United Kingdom. Their fields of study included business management, economics, marketing, tourism, events management and logistics. Participants in Study 1 were randomly allocated into two different groups: Control 1 (n = 105) and Human coaching (105). For Study 2, participants were allocated into Control 2 (n = 134) and AI coaching (n = 134) groups. In total over the two studies, 327 participants successfully submitted data over all eight time-points which were used for the data analysis. See Table 2 for group numbers, demographic distribution, and mean scores of the dependent variable used in this study.

Table 2. Goal attainment means of the four participant groups across the eight time-points.

|  | Control 1 | Human coaching | Control 2 | AI coaching |
|---|---|---|---|---|
| *n* | 85 | 94 | 73 | 75 |
| **Gender** | 77% female | 64% female | 60% female | 54% female |
| **Age** | 23.04 (18–53) | 22.56 (18–51) | 23. 81 (18–46) | 21.57 (18–48) |
| **Goal attainment 1** | 25.95 (12.50) | 27.66 (11.65) | 27.22 (12.78) | 26.06 (12.17) |
| **Goal attainment 2** | 27.68 (14.54) | 27.03 (11.22) | 27.03 (12.30) | 24.89 (12.19) |
| **Goal attainment 3** | 29.31 (14.59) | 31.62 (11.56) | 30.53 (13.25) | 30.05 (11.31) |
| **Goal attainment 4** | 31.26 (15.42) | 34.64 (13.16) | 32.56 (14.60) | 33.04 (13.58) |
| **Goal attainment 5** | 32.25 (14.93) | 35.93 (13.78) | 31.35 (14.68) | 34.10 (12.46) |
| **Goal attainment 6** | 33 (14.95) | 37.32 (13.71) | 32.30 (15.27) | 35.64 (16.15) |
| **Goal attainment 7** | 33.47 (14.79) | 41.29 (14.58) | 34.59 (15.04) | 39.36 (17.18) |
| **Goal attainment 8** | 37.74 (17.49) | 41.15 (15.43) | 35.01 (16.04) | 41.11 (17.25) |

*Note.* The table shows number of participants, distribution of females, mean age (min-max), mean scores of goal attainment and (standard deviation within brackets).

Capturing the placebo effect in research requires subjects to not be aware whether they receive treatment or not [77]. In these two studies we were unable to investigate the placebo effect as coached students were given access to a human coach or the AI coach (Vici) almost immediately after the start of the experiment. The students in the control group were also aware of not having access to human or AI coaching. To address outcomes expectancy [78], the control groups received a fact sheet that provided information about goal attainment, psychological wellbeing, resilience and perceived stress at the start of the trails. They were also asked to think of and specify goals they wanted to achieve over the next ten months. In RCT studies there is a danger of a nocebo effect where participants in the control group have negative outcome expectations because they are aware of not receiving the intervention. In the present study we believe this was managed based on the research of Colloca et al. [79], who had found that a higher number of exposures to trial conditioning correlated to longer duration of nocebo responses. In our study, control group participants were only "conditioned" (made aware of not receiving coaching and given information sheets) at the start of the trials. The relatively long duration of the trials (10 months) likely also helped to diminish the nocebo effect.

## Procedure

Participants conducted a survey over eight time-points using an online survey platform. The first survey was a baseline survey before the participants had been allocated to any of the conditions. The baseline survey collected demographic data, dependent variables and the participants were asked to specify two goals that they aimed to work on over the phase of the project. These goals were supposed to be something challenging that was either new or something that has been difficult for them to achieve in the past. After the baseline survey had been completed, the participants completed a monthly survey for six months where the participants were asked to rate the success and the difficulty of their two goals. The monthly survey was distributed for Control group 1, Control group 2, and the AI coaching group. The participants of the Human coaching group received their survey after they had conducted their monthly coaching session. The eighth survey was distributed to all the participants three months after the last monthly survey.

Each student had unique login details to their survey, allowing the participants to be reminded of their goals and allowing the administrators of the survey to send out reminders to the participants when submissions were missing.

## Human coaching

Students were coached by professional coaches trained and qualified in a relational model of coaching by a UK-based institution. Their qualifications included at least the Ashridge Executive Coach Accreditation and the European Individual Award (EIA) by the European Mentoring & Coaching Council at Senior Practitioner level. The 105 coaches were on average 50 years old and had at least three years of business coaching experience. Coaches and students were matched randomly with each coach assigned to one student only. Students had six one-hour coaching sessions over a period of six months, one session per month. All sessions were conducted via Skype. There was no prescription on the topic or content of the coaching sessions. Coaches and participants had complete freedom to decide how they wanted to use the session, which topics and goals they wanted to set and pursue and which homework tasks between sessions needed to be completed. All participants had to participate in all six coaching sessions to remain part of the study.

## AI coaching

Applying the principle of narrow AI to coaching suggests the creation of a form of artificial narrow intelligence (ANI) that can perform one specific coaching task, rather than an attempt to create a machine replica of a human coach. The AI coach used in this study, Coach Vici was based on expert system (ANI) principles using chatbot technology. The sole purpose of the chatbot was to help participants with goal attainment. Expert systems are considered a form of narrow AI and are described as complex software programs based on specialized knowledge, able to provide acceptable solutions to individual problems in a narrow topic area [80, 81]. Chatbots in turn are computer programs that interact with users via natural language either through text, voice, or both [82].

Vici is a custom-developed text-based chatbot deployed on the Telegram instant messaging platform. The chatbot was developed using the Designing AI Coach (DAIC) framework that recommends merging aspects of strong human coaching relationship with chatbot design best practices and using proven, evidence-based coaching theories as foundation [4]. In line with these recommendations, Vici was designed to facilitate goal attainment according to goal theory [55]. Vici had two types of text-based conversations with users. In the first type of conversation, Vici helped users to specify realistic goals by questioning them on the importance, feasibility and impact of their stated goals. Vici then helped users to commit to achievable actions that would help them reach their goals. In the second type of conversation, users would check in with Vici to report on their goal and action progress, reflect on obstacles that prevented them from progressing and changing their actions plans if necessary. These conversations assisted users to monitor the progress of their goals and actions. Vici also helped users to distinguish between proximal (< 6 months) and distal (> 6 months) goals [83]. Vici was available 24/7 to the experimental group and they could use it as often as they wanted, but at least once a month. A detailed analysis of the AI coach usages is presented in the Results. Fig 1 shows examples of interactions with Coach Vici.

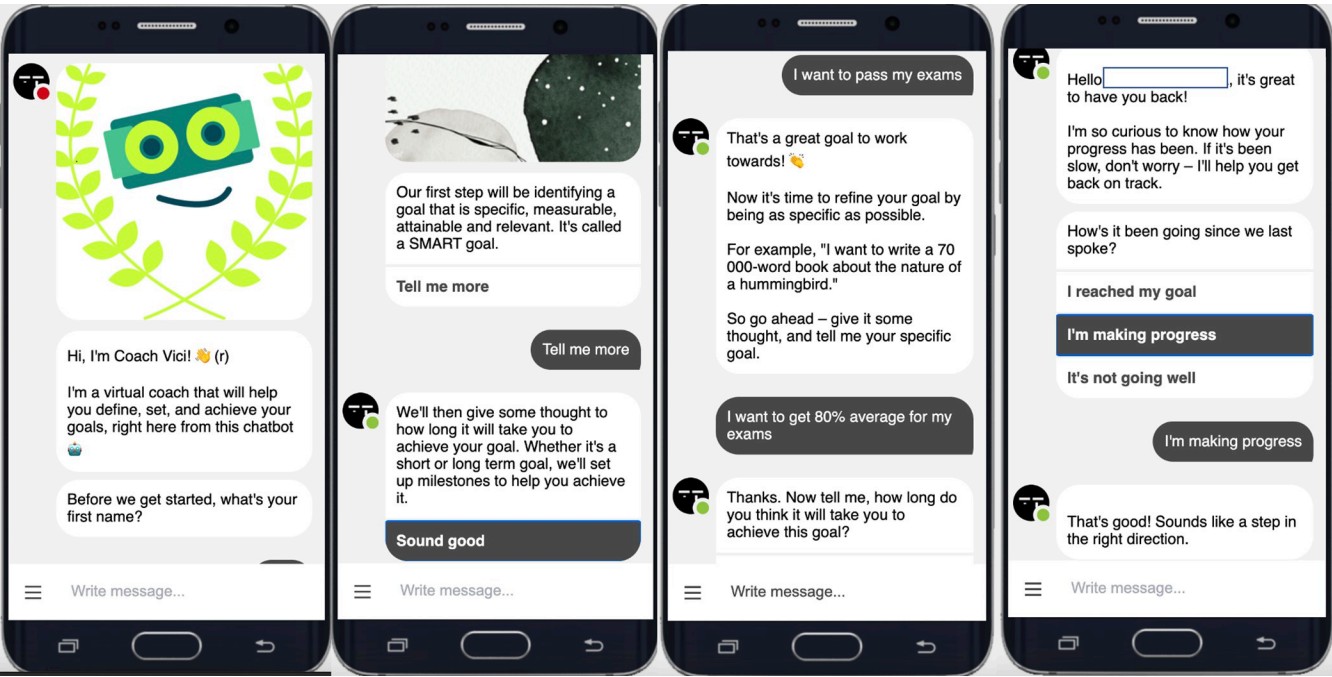

**Fig 1.**

## Measures

### Goal attainment

Grant et al. [67] developed a goal attainment measure which was adapted for the purpose of this study. The goal attainment measure contained self-reported scores of how successful the participants perceived they had been in achieving their goals and how difficult they perceived their goals. The successful score was measured on 11 points, where each point represented 0%, 10%, 20%, etc. up to 100%. The difficulty score was measured using a seven-point Likert scale ranging from 'Very easy' to 'Very difficult'.

The overall goal attainment score was then calculated by multiplying the success and the difficulty score for each goal separately and dividing the scores for the two different goals to create an average goal attainment score.

## Results

To assess the implications of coaching on goal attainment, a 4x8 Mixed Factorial ANOVA was conducted using the four different groups (Control 1, Control 2, Human coach group and AI coach group) as grouping variable and their eight self-reported measures of goal attainment as dependent variables. A power-analysis using G*Power 3.1.9.7 [84] was conducted to determine the effect size required to identify a statistically significant interaction between four groups over eight time-points. A Mixed Factorial ANOVA with a within-between interaction of 327 participants, a power of 0.95 and alpha level of 0.05 indicated that effect of the coaching intervention would have to be above $\eta_p^2 = .014$ to identify a significant interaction.

The Mixed Factorial ANOVA indicated a statistically significant interaction of group and time, $f(13.18, 1296.36) = 2.35$, $p = .004$, $\eta_p^2 = .023$. To break down this interaction, the development of goal attainment was first analysed within each group using separate Repeated Measures ANOVA over the eight time-points. The Repeated Measures ANOVA indicated that all groups had a significant development of Goal Attainment over time. Both control groups behaved similar, and the trend of the development remained similar over the two time-points. The first control group reported and effect size of $\eta_p^2 = .16$, $p < .001$ and the second control group showed an effect size of $\eta_p^2 = .11$, $p < .001$. Both control groups significantly developed their goal attainment from baseline to time-point 4 ($p = .005$ and $p = .003$ respectively), but kept it at a stable level for the remaining time-points.

Furthermore, the effects size over the eight time-points of the two experiment groups were remarkably higher compared to those of the control groups (Human coach group = $\eta_p^2 = .265$, $p < .001$ and AI coach group = $\eta_p^2 = .269$, $p < .001$). The first significant development of goal attainment was identified at time-point 4 for both experiment groups ($p = .01$ and $p = .004$ respectively), but the development kept significantly increasing over time-point 7 and time-point 8 for both experiment groups.

Bonferroni corrected multiple comparisons were used to test the difference between each of the four groups at each time-point. The tests indicated that a significant difference between the Human coach and Control group 2 could be identified at time-point 5 where the human experiment group had a significant higher goal attainment score ($p = .047$) than Control group 2. This effect increased with time and both the Human coach group, and the AI coach group showed significant effects compared to those of the control groups at time-point 7 (Human coach group–Control 1, $p = .002$; Human coach group–Control 2, $p = .008$; AI coach group–Control 1, $= .022$; AI coach group–Control 2 $= .048$). However, the goal attainment of the two experiment groups never significantly differed between each other throughout the experiment.

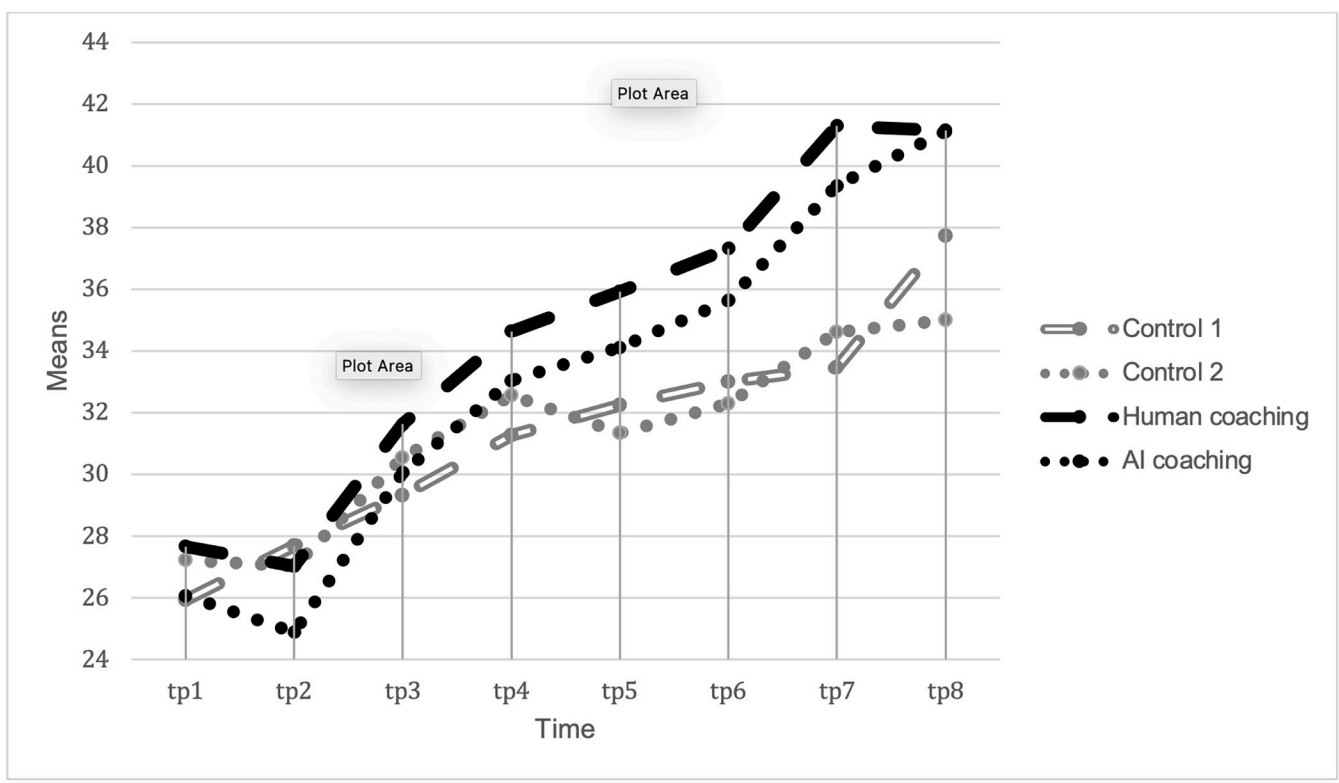

**Fig 2.**

The effect size of $\eta_\rho^2$ = .023 which was higher than the critical effect size ($> \eta_p^2$ = .014) according to the power analysis indicated that the sample size for this study was sufficient. These findings further indicate that the participants successfully increased their goal attainment over the time of the study. As shown in Fig 2, receiving coaching had a positive impact on the development of goal attainment for the participants, but both formats of coaching appeared to have had very similar effect.

We also analyzed the usage of the AI coach in terms of usage frequency of the chatbot to identify any potential within-group differences in Study 2. We were able to identify a significant difference in development of goal attainment when splitting the frequency of usage into two equal groups based on their median usage (6 AI coaching sessions, $t$ (73) = -2.24, $p$ = 0.028, $d$ = 0.52. The lower usage group had an average increase on goal attainment of 17.62 (sd = 32.50) compared to 37.62 (sd = 34.16) in the higher usage group. This suggests that more frequent use of the AI coach led to higher goal attainment.

To understand the nature of goals across the two studies and four groups, two of the authors independently analysed the first goal in both studies to assess the theme of the goal, the type of outcome (concrete or vague goal) and whether the goal was proximal (<6 months) or distal (>6 months). The inter-rater reliability of the analysis indicated a very high similarity between the reviewers on all three categories, with Cohen's kappa of $\kappa$ = .95, p < .001 for the theme of the goal, $\kappa$ = .94 for the outcome and $\kappa$ = .91, p < .001 for the timeline of the goal.

The themes of the goals that were identified related to the participants' studies (38%), self-development (22%), career (18%), health and wellbeing (16%), other (2%), finances (2%) family (1%) and property or car (1%). Most of the goals were concrete and measurable (60%), for example, "To gain overall mark of 75% in study year 1" and 58% of the goals were long-term

focused (>6 months). Furthermore, the proportion of themes, type or proximal within the four different groups did not significantly differ among each other.

The type of outcome and the proximity of the goals were added as covariates into the model, but neither had a significant impact on the development of goal attainment ((outcome, $f(7, 2037)$ = 1.46, $p > .05$, $\eta_p^2 = .005$) and proximity, $f(7, 2037) = 2.06$, $p > .05$, $\eta_p^2 = .007$)). The themes were analysed separately due to the large variety of the themes, but no significant differences between any of the themes were found on the development of goal attainment, $f(7, 334) = .80$, $p > .05$, $\eta_p^2 = .017$). These findings indicate that the individual differences in the goals among the participants did not impact the development of goal attainment over time in this study.

## Discussion

This paper investigates the performance of an AI chatbot coach relative to human coaches in terms of client goal attainment by comparing two longitudinal RCT studies, the second being a replication study of the first. In both studies the experimental groups who had received either human coaching (Study 1) or AI coaching (Study 2) had significantly higher goal attainment than the control groups. A surprising result is that the AI coach rivalled the human coaches in participant goal attainment with a similar outcome at the end of the study after ten months (Human coach group = $\eta_\rho^2 = .265$, $p < .001$ and AI coach group = $\eta_\rho^2 = .269$, $p < .001$). Using goal theory, we attempt to explain this result and we discuss three important implications: (i) the possibility of democratizing coaching; (ii) the way AI coaching could enlarge the need for human coaches; and (iii) a warning to coaches to enhance their coaching praxis.

Goal theory states that there is a higher level of goal attainment if goals are clear, there is buy-in from the onset, regular feedback is provided on progress, and the goals are sufficiently challenging and not too complex. Practically this translates to writing down goals, having specific time frames associated with a goal, measuring progress and making a commitment to someone about completing the goal [55]. Most coaching approaches have at their core the notion of goal attainment. In fact, this is what sets coaching apart from other helping professions [64] and explains why human coaches were able to help participants increase their goal attainment. Goal theory and how it is typically practised by human coaches was used to create the AI coach, Vici. The AI coach helped participants write down their goals by typing it into the application, asked questions to test for the feasibility and level of realism of the goals, and went even further by helping participants create an action plan to reach their goals. Human coaches would be able to engage in a more complex and nuanced discussion about goals and one would therefore expect human coaches to outperform the AI coach who employs a scripted conversation. Although the AI coach lacked nuanced intelligence, it had an advantage over human coaches due to the rigorous and consistent way it executed goal theory. The human coach could decide which aspects of goal theory they implemented in each session and in fact, some of the coaches may have not been well versed in goal theory. For example, in the human coaching sessions the coaches may occasionally have forgotten to ask about goal progress or potentially did not keep an explicit record of goal progress as it is up to each coach to decide how to manage the coaching intervention. The AI coach, however, was programmed to always enquire about goal progress and keep a record for reference to share with the participant. Furthermore, human coaches may have asked participants to verbalize their goals instead of writing them down, whereas the AI coach required the participants to type their goals into the application as writing down a goal has been shown to increase goal attainment [55]. It seems therefore that the rigor and mechanistic execution of goal theory by the AI coach and its inability to deviate from a set process (which could in fact detract human coaches) compensated for its lack of human intelligence.

The results (Fig 2) show that the AI coach trailed the human coaches slightly throughout the eight measurements up to the last time-point (T8). Measures were taken monthly between T1 and T7 with a final follow-up measure (T8) after three months. Between T7 and T8, human coach participants did not receive any more coaching, whereas the AI coaching participants were free to keep using the AI coach. This could explain why goal attainment of human coaching participants declined between T7 and T8, but kept increasing for the AI coach group to ultimately equal the human coach group. The convenience and constant availability of the AI coach probably assisted in its performance relative to human coaches.

An important predictor of coaching success is the readiness of the coachee [85]. Due to the randomised nature of this study, we can assume that in both studies participants were equally open and ready for coaching. Being perceptive to coaching relates to a person's state at a particular time of day such as their energy levels, mental alertness, and general physical state. In the human coaching group, sessions were scheduled in advance and because two people are involved in the logistics, one can assume that at times appointments were honoured despite the coach or participant not being in an optimal state for the engagement, potentially negatively affecting the efficacy of that session. In the AI coaching group, the participants had complete freedom to decide when to have a conversation with the chatbot, which may have contributed to a more optimal engagement. While we did not explicitly measure these variables, we suggest that this convenience factor may have helped the AI coach to perform well compared to human coaches. Additionally, the AI coach was available 24/7 and participants could use it as often as they chose.

The results indicate that participants who used the AI coach more often had higher goal attainment. Human coaching is expensive and therefore participants in Study 1 only had one session per month. There was no extra cost associated with additional AI coaching sessions. This underscores two of the main advantages of AI coaching–its scalability and cost effectiveness compared to human coaching. The superior availability and use of the AI coach compared to the human coaches could therefore also explain why the AI coaching group performed so well relative to the human coaching group.

The implications of these results are three-fold. Firstly and most importantly, this presents the possibility of democratization of coaching. A number of coaching efficacy meta-studies have shown coaching to be effective in helping people develop, grow and achieve their goals. Coaching is however reserved for a select few due to its cost and the availability of coaches, especially on low-income geographies such as Africa. Even in organizations, individual coaching is usually reserved for the managers and senior leaders. The results from this comparative study suggest that AI coaching, when implemented to have a specific focus in line with the current capabilities of narrow AI, is an affordable and scalable alternative to certain aspects of human coaching. The benefits of coaching could therefore be made available to many more people than is currently the case.

The second implication relates to the coaching industry. Many people are concerned that AI threatens their job security [86]. Coaches may therefore rightly be concerned that AI coaches, such as Vici, pose a threat to their livelihood. The opposite may in fact be true. If AI can help democratize coaching, more first-time users of coaching services would be exposed to the benefits of coaching. Due to the limited abilities of AI, at some point users of AI coaching services may have the need for more advanced and intelligent human coaching. We believe this broadening awareness of and exposure to coaching through AI could in fact create more opportunities for human coaches. Human coaches should view AI coaching as an opportunity, not a threat, in line with the findings of a recent study [74].

The final implication relates to coaches and their praxis. The efficacy of the AI coach in this study suggests that coaches who operate at a low level of coach maturity [73, 75] could be

replaced by AI coaches. Therefore, human coaches need to evaluate their coach maturity and invest the necessary resources to improve their coaching knowledge and skills to ensure that they offer their clients a valuable and relevant service. Humans currently and for the foreseeable future will outperform AI in terms of context awareness, transference of learning and higher order complex sense-making [17]. Coaches should ensure that they embody these uniquely human forms of intelligence in their coaching praxis.

## Limitations and future research

Participants in both studies were undergraduate students. This implies that the results may not generalise to other populations; however, the effects observed are still valid given that similar groups of participants were used in both studies. Measurements were performed by means of self-scores by participants, which may introduce the possibility of self-score bias. These limitations are offset to some degree by the relatively large sample size and the longitudinal, RCT research design.

In terms of future research, other narrowly focused AI coaches, who specialise in one specific coaching aspect such as wellness, self-awareness or emotional intelligence, should be created and used in a replication study similar to this goal-attainment AI coach. This would help us understand what other coaching aspects can be automated. Should some of these other coaching aspects yield positive results in an AI implementation, the possibility of creating a composite AI coach consisting of an amalgamation of these narrow AI capabilities should be researched. While general AI is not yet possible, perhaps the sum of numerous narrow AI coaching capabilities could create a synergetic AI coaching effect.

## Conclusion

Uniquely human characteristics such as emotional intelligence and empathy allows human coaches to build bonds with their clients that no AI can currently rival. This comparison study however shows that AI coaches that focus on a narrow aspect of coaching and are based on fundamental, proven theories may very well rival human coaches in that specific coaching aspect. While AI coaches will not out-perform human coaches as a whole any time soon, these specific applications of coaching could democratize coaching and make its benefits available to a much wider audience, while at the same time potentially growing the demand for human coaches through exposing more people to the benefits of coaching.

## Supporting information

**S1 Data.**
(XLSX)

**S1 File.**
(DOCX)

## Author Contributions

**Conceptualization:** Nicky Terblanche, Joanna Molyn, Erik de Haan.

**Data curation:** Joanna Molyn, Viktor O. Nilsson.

**Formal analysis:** Viktor O. Nilsson.

**Investigation:** Nicky Terblanche.

**Project administration:** Joanna Molyn.

**Software:** Nicky Terblanche.

**Writing – original draft:** Nicky Terblanche.

**Writing – review & editing:** Joanna Molyn, Erik de Haan, Viktor O. Nilsson.

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
