## [Decision Letter · Decision Letter 0]

13 Dec 2021

PONE-D-21-31265Comparing Artificial Intelligence and Human Coaching Goal Attainment EfficacyPLOS ONE

Dear Dr. Terblanche,

Thank you for submitting your manuscript to PLOS ONE. After careful consideration, we feel that it has merit but does not fully meet PLOS ONE’s publication criteria as it currently stands. Therefore, we invite you to submit a revised version of the manuscript that addresses the points raised during the review process.

 Two Reviewers evaluated the manuscript. Opinions were inconsistent, and not all Reviewers were supportive of further consideration of the contribution. I encourage Authors to take into account Reviewers' concerns, especially those related to clarity in methodology and depth of theoretical background - importantly, while the background is rich in info for what regards coaching, it is very poor for what regards human-Artificial Intelligence interaction literature.  Authors should take into account that another round of revisions will be probably needed, and that acceptance of the final manuscript could not be guaranteed. 

We look forward to receiving your revised manuscript.

Kind regards,

Stefano Triberti, Ph.D.

Academic Editor

PLOS ONE

Journal Requirements:

Reviewers' comments:

Reviewer's Responses to Questions

**Comments to the Author**

1. Is the manuscript technically sound, and do the data support the conclusions?

Reviewer #1: Yes

Reviewer #2: Partly

2. Has the statistical analysis been performed appropriately and rigorously? 

Reviewer #1: Yes

Reviewer #2: No

3. Have the authors made all data underlying the findings in their manuscript fully available?

Reviewer #1: Yes

Reviewer #2: No

4. Is the manuscript presented in an intelligible fashion and written in standard English?

Reviewer #1: No

Reviewer #2: Yes

5. Review Comments to the Author

Reviewer #1: 1. More emphasis should be put on the procedure enacted to counter the placebo effect. By informing the control group of the coaching benefits, we may incur a nocebo effect. How do you account for that?

2. The meeting with the human coach is once per month. However, nothing is said for the chatbot that can be consulted 24/7. Is there a positive correlation between goal attainment and frequency of usage?

3. There is a typo in "An interesting notion it that ..."

Reviewer #2: This manuscript compares two equivalent longitudinal RCT studies that measured the increase in clients' goal attainment as a result of receiving coaching over a 10-month period.

In my opinion, the manuscript has the potential to make a nice contribution to the study of Artificial Intelligence, however, it needs radical implementations and, in this current form, it is not acceptable for publication to PLOSONE Journal.

I reported below some suggestions for future submission:

Despite the introduction include some well-written sections, it could be interesting to understand in which contexts AI is used today.

The authors cited articles that are currently under review. For future submission, I discourage citing articles that it is not yet published (unless the articles are “in press”).

The authors stated that AI will remain unconscious machines that can at best support humans in complex, specific tasks.

However, we know that is expected that AI will participate more and more in decision-making processes. In this sense could be interesting to discuss in the paper the effect of AI on human relationships. For example, this article explores the effect of AI on health decision-making. Triberti, S., Durosini, I., & Pravettoni, G. (2020). A “Third Wheel” Effect in Health Decision Making Involving Artificial Entities: A Psychological Perspective. Frontiers in public health, 8, 117.

The authors describe the Goal theory. It is important to learn more about the use of goal theory in coaching, also presenting studies on its efficacy and clarifying its relationship with AI in coaching.

Additionally, I have several doubts about the experimental design:

- The authors declared that they conducted an RCT. However, it is not explained the criteria and the research protocol of the research design (es. did the authors use the CONSORT checklist?). More information is needed.

- It is not clear why the authors carried out the data collection in different years and why they compared two different control groups. If there is a reason for this, it must be specified in the text.

- It is not clear how the meetings with human coaching were carried out. What topics were covered during the meetings? Did all participants participate in all meetings? What requests were made to the participants after the meeting? It is necessary to better clarify how it was carried out, also using tables or figures.

- The same with the AI chatbot. It is not clear what content is offered. It could be useful to present examples.

- The chatbot was also searchable 24/7 by users. This is a big difference from the experimental group with human coaching. This difference could have an impact on the results.

- It is also useful to report how many times participants have consulted AI coaching (if available) and understand how this can affect the results (e.g., use this variable as a covariate).

- Additionally, people have many different goals. The authors describe very briefly this aspect in the manuscript. I think it would be useful to describe accurately in the paper the goals classification and consider them in the statistical analyses.

Finally, in the discussion, it is important to accurately discuss the practical implication of the results. It would also be useful to underline how human characteristics (e.g., emotional intelligence) make the human figure irreplaceable in the relationship with clients.

For these reasons, despite I consider the study itself an interesting contribution to the study of Artificial Intelligence, the manuscript is not acceptable for publication in its current form.

6. PLOS authors have the option to publish the peer review history of their article (what does this mean?). If published, this will include your full peer review and any attached files.

Reviewer #1: No

Reviewer #2: No

---

## [Author Response · Author response to Decision Letter 0]

12 May 2022

PLOS ONE

Author’s response to reviewers’ comments for:

PONE-D-21-31265: Comparing Artificial Intelligence and Human Coaching Goal Attainment Efficacy

Dear Editor and Reviewers

Thank you kindly for affording us the opportunity to review our paper. We found the comments and suggestions very helpful and hope we have addressed your concerns in full.

This is very important, first-of-kind research for the coaching community – researchers, practitioners and purchasers of coaching services – as it may pave the way for more mainstream use of AI in the coaching. We would therefore be honoured to share this with the world through your respected and high impact journal.

Thank you

# Editor’s comments Response from author

1 Clarity in methodology 

RESPONSE: The methodology has been amended to clarify the questions of the reviewers. Please refer to detailed comments below.

2 Depth of theoretical background - importantly, while the background is rich in info for what regards coaching, it is very poor for what regards human-Artificial Intelligence interaction literature 

RESPONSE: The literature review was extensively edited to include two new sections on AI and coaching:

Human-AI interactions and relationships in the context of Coaching

Potential benefits and ethical challenges in AI coaching

3 Please ensure that your manuscript meets PLOS ONE's style requirements, including those for file naming. 

RESPONSE:Referencing was changed from APA to Vancouver and other stylistic changes also made to comply with PLOS ONE requirements 

4 We note that you have indicated that data from this study are available upon request. PLOS only allows data to be available upon request if there are legal or ethical restrictions on sharing data publicly. 

RESPONSE: Anonymised data is included in the submission as Supporting Information files

5 In your revised cover letter, please address the following prompts:

If there are no restrictions, please upload the minimal anonymized data set necessary to replicate your study findings as either Supporting Information files or to a stable, public repository and provide us with the relevant URLs, DOIs, or accession numbers. For a list of acceptable repositories, please see http://journals.plos.org/plosone/s/data-availability#loc-recommended-repositories.

RESPONSE: See #4 above

6 Please include your full ethics statement in the ‘Methods’ section of your manuscript file. In your statement, please include the full name of the IRB or ethics committee who approved or waived your study, as well as whether or not you obtained informed written or verbal consent. If consent was waived for your study, please include this information in your statement as well. RESPONSE: Added anonymous reference to a UK University (for blind review)

The research was approved by the ethics committee of a London-based University, project reference UREC/19.1.5.6. Written informed consent was obtained from all participants in both studies as per the requirements of the ethics approval.

# Reviewer #1 comments Response from author

1 More emphasis should be put on the procedure enacted to counter the placebo effect. By informing the control group of the coaching benefits, we may incur a nocebo effect. How do you account for that? 

RESPONSE: Addded the following in Methodology

In RCT studies there is danger of a nocebo effect where participants in the control group have negative outcome expectations because they are aware of not receiving the intervention. In the present study we believe this was managed based on the research of Colloca et al. (2010). They found that a higher number of exposures to trial conditioning correlated to longer duration of nocebo responses. In our study control group participants were only “conditioned” (made aware of not receiving coaching and given information sheets) at the start of the trials. The relatively long duration of the trials (10 months) likely also helped to diminish the nocebo effect. 

2 The meeting with the human coach is once per month. However, nothing is said for the chatbot that can be consulted 24/7. Is there a positive correlation between goal attainment and frequency of usage? 

RESPONSE: Yes there is. Added the following in the Results section 

We also analyzed the usage of the AI coach in terms of how many times the AI coaching chatbot was used to identify any potential within-group differences in the second study. We were able to identify a significant difference in development of Goal Attainment when splitting the frequency of usage into two equal groups based on their median usage (6 AI coaching sessions, t (73) = -2.24, p = 0.028, d = 0.52. The lower usage group had an average increase on Goal Attainment of 17.62 (sd = 32.50) compared to 37.62 (sd = 34.16) in the higher usage group. This suggests that more frequent use of the AI coach led to higher goal attainment. 

Also added this to the Discussion section:

Additionally, the AI coach was available 24/7 and participants could use it as often as they chose. The results indicate that participants who used the AI coach more often had higher goal attainment. Human coaching is expensive and therefore participants in study 1 only had one session per month. There was no extra cost associated with additional AI coaching sessions. This underscores one of the main advantages of AI coaching – it’s scalability and cost effectiveness compared to human coaching. The superior availability and use of the AI coach compared to the human coaches could therefore also explain why the AI coaching group performed so well relative to the human coaching group.

3 There is a typo in "An interesting notion it that ..." 

RESPONSE: This sentence has been reworded

# Reviewer #2 comments Response from author

1 Despite the introduction include some well-written sections, it could be interesting to understand in which contexts AI is used today.

RESPONSE: The literature review was extensively edited to include two new sections on AI:

Human-AI interactions and relationships in the context of Coaching

Potential benefits and ethical challenges in AI coaching

2 The authors cited articles that are currently under review. For future submission, I discourage citing articles that it is not yet published (unless the articles are “in press”). 

RESPONSE: Thank you for this remark. These references have been removed.

3 The authors stated that AI will remain unconscious machines that can at best support humans in complex, specific tasks.

However, we know that is expected that AI will participate more and more in decision-making processes. In this sense could be interesting to discuss in the paper the effect of AI on human relationships. For example, this article explores the effect of AI on health decision-making. Triberti, S., Durosini, I., & Pravettoni, G. (2020). A “Third Wheel” Effect in Health Decision Making Involving Artificial Entities: A Psychological Perspective. Frontiers in public health, 8, 117. 

RESPONSE: Thank you for the reference to this helpful paper. It was included in the new expanded literature review that also covers various aspects of AI in these two sections: 

Human-AI interactions and relationships in the context of Coaching

Potential benefits and ethical challenges in AI coaching

4 The authors describe the Goal theory. It is important to learn more about the use of goal theory in coaching, also presenting studies on its efficacy and clarifying its relationship with AI in coaching. 

RESPONSE: More literature added about efficacy studies of coaching and goal attainment and links to AI as follows:

Several empirical studies have shown that coach is effective in goal attainment. In a RCT study Grant et al. (2009) found that four coaching sessions over a 10-week period led the intervention group to a significantly higher level of goal attainment compared to the control group. Zimmerman and Antoni (2018) analyzed 33 coaching dyads using longitudinal multilevel analyses and found that clients experienced increased goal attainment. Losch et al. (2016) compared individual coaching, self-coaching and group training and found that individual coaching was effective and superior in helping leaders achieve their goals. 

We conclude that goal theory is well understood, pragmatic, intrinsic in coaching and that coaching has been shown to improve goal attainment. It is therefore an ideal foundation theory for use in a narrow AI application.

5 The authors declared that they conducted an RCT. However, it is not explained the criteria and the research protocol of the research design (es. did the authors use the CONSORT checklist?). More information is needed. 

RESPONSE: Added the following under Methodology

The CONSORT guidelines for RCT research design were followed and adhered to as far as possible (Schulz, 2010).

6 It is not clear why the authors carried out the data collection in different years and why they compared two different control groups. If there is a reason for this, it must be specified in the text. 

RESPONSE: Added the following under Methodology:

The second study was conducted after the first study because the AI coach used was only created in 2019 after the completion of the first study.

7 It is not clear how the meetings with human coaching were carried out. What topics were covered during the meetings? Did all participants participate in all meetings? What requests were made to the participants after the meeting? It is necessary to better clarify how it was carried out, also using tables or figures. 

RESPONSE: Added more detail under “Human coaching” in the methodology section

Coaches and participants had complete freedom to decide how they wanted to use the session, which topics and goals they wanted to set and pursue and which homework tasks between sessions needed to be completed. All participants had to participate in all six coaching sessions to remain part of the study.

8 The same with the AI chatbot. It is not clear what content is offered. It could be useful to present examples. 

RESPONSE: Added four screen shots (figure 1) to illustrated Coach Vici’s conversation

9 The chatbot was also searchable 24/7 by users. This is a big difference from the experimental group with human coaching. This difference could have an impact on the results. 

RESPONSE: Added the following in the Results section 

We also analyzed the usage of the AI coach in terms of how many times the AI coaching chatbot was used to identify any potential within-group differences in the second study. We were able to identify a significant difference in development of Goal Attainment when splitting the frequency of usage into two equal groups based on their median usage (6 AI coaching sessions, t (73) = -2.24, p = 0.028, d = 0.52. The lower usage group had an average increase on Goal Attainment of 17.62 (sd = 32.50) compared to 37.62 (sd = 34.16) in the higher usage group. This suggests that more frequent use of the AI coach led to higher goal attainment. 

Also added this to the Discussion section:

Additionally, the AI coach was available 24/7 and participants could use it as often as they chose. The results indicate that participants who used the AI coach more often had higher goal attainment. Human coaching is expensive and therefore participants in study 1 only had one session per month. There was no extra cost associated with additional AI coaching sessions. This underscores one of the main advantages of AI coaching – it’s scalability and cost effectiveness compared to human coaching. The superior availbailty and use of the AI coach compared to the human coaches could therefore also explain why the AI coaching group performed so well relative to the human coaching group.

10 It is also useful to report how many times participants have consulted AI coaching (if available) and understand how this can affect the results (e.g., use this variable as a covariate). See point above

11 Additionally, people have many different goals. The authors describe very briefly this aspect in the manuscript. I think it would be useful to describe accurately in the paper the goals classification and consider them in the statistical analyses. 

RESPONSE: Added more stats regarding goals in the Results section:

To understand the nature of goals across the two studies and four groups, two of the authors independently analysed the first goal in both studies to assess the theme of the goal, the type of outcome (concrete or vague goal) and whether the goal was proximal (<6 months) or distal (>6 months). The inter-rater reliability of the analysis indicated a very high similarity between the reviewers on all three categories, with Cohen’s kappa of κ = .95, p < .001 for the theme of the goal, κ = .94 for the outcome and κ = .91, p <.001 for the timeline of the goal. 

The themes of the goals that were identified relate to the participants’ studies (38%), self-development (22%), career related (18%), health and well-being (16%), other (2 %), financial related (2%) family related goals (1%) and property or car related (1%). Most of the goals were concrete and measurable (60%), for example, “To gain overall mark of 75% in study year 1” and 58% of the goals were long term focused (> 6 months). Furthermore, the proportion of themes, type or proximal within the four different groups did not significantly differ between each other.

The type of outcome and the proximity of the goals were added as covariates into the model but neither had a significant impact on the development of goal attainment ((outcome, f (7, 2037) = 1.46, p > .05, ηp2 = .005) and proximity, f (7, 2037) = 2.06, p > .05, ηp2 = .007)). The themes were analysed separately due to large variety of the themes but no significant differences between any of the themes were found on the development of goal attainment, f (7, 334) = .80, p > .05, ηp2 = .017). These findings indicate that the individual differences in the goals among the participants did not impact the development of goal attainment over time in this study. 

12 Finally, in the discussion, it is important to accurately discuss the practical implication of the results. It would also be useful to underline how human characteristics (e.g., emotional intelligence) make the human figure irreplaceable in the relationship with clients. 

RESPONSE: Changed the Conclusion section to include this argument. However I want to emphasise that AI coaching can still be effective even without this human connection. This I believe is the novel contribution of this research study:

Uniquely human characteristics such as emotional intelligence and empathy allows human coaches to build bonds with their clients that no AI can currently rival. This comparison study however shows that AI coaches that focus on a narrow aspect of coaching and are based on fundamental, proven theories may very well rival human coaches in that specific coaching aspect. While AI coaches will not out-perform human coaches as a whole any time soon, these specific applications of coaching could democratize coaching and make its benefits available to a much wider audience, while at the same time potentially growing the demand for human coaches through exposing more people to the benefits of coaching.

---

## [Editor Report · Decision Letter 1]

8 Jun 2022

Comparing Artificial Intelligence and Human Coaching Goal Attainment Efficacy

PONE-D-21-31265R1

Dear Dr. Terblanche,

We’re pleased to inform you that your manuscript has been judged scientifically suitable for publication and will be formally accepted for publication once it meets all outstanding technical requirements.

Kind regards,

Stefano Triberti, Ph.D.

Academic Editor

PLOS ONE
---

## [Editor Report · Acceptance letter]

10 Jun 2022

PONE-D-21-31265R1 

Comparing artificial intelligence and human coaching goal attainment efficacy 

Dear Dr. Terblanche:

I'm pleased to inform you that your manuscript has been deemed suitable for publication in PLOS ONE. Congratulations! Your manuscript is now with our production department. 

Kind regards, 

on behalf of

Dr. Stefano Triberti 

Academic Editor

PLOS ONE